# Genetic diversity of whitefly species of the *Bemisia tabaci* Gennadius (Hemiptera: Aleyrodidae) species complex, associated with vegetable crops in Côte d'Ivoire

Anthelme-Jocelin N'cho[1,2,3,4]*, Koutoua Seka[2], Kouamé Patrice Assiri[2], Christophe Simiand[1], Daniel H. Otron[1], Germain Ochou[3], Kouassi Arthur Jocelin Konan[3,5], Marie-France Kouadio[3,5], Lassina Fondio[3], Hortense Atta Diallo[2], Thibaud Martin[4,5], Hélène Delatte[6]*

**1** Cirad, UMR PVBMT, Saint-Pierre, La Réunion, France, **2** Nangui Abrogoua University, Abidjan, Côte d'Ivoire, **3** National Center for Agricultural Research (CNRA), Bouaké, Côte d'Ivoire, **4** University of Montpellier, Cirad, UR Hortsys, Montpellier, France, **5** Felix Houphouet Boigny University of Cocody, Abidjan, Côte d'Ivoire, **6** Cirad, UMR PVBMT, Antananarivo, Madagascar

* anthelmejocelinncho@gmail.com (AJN); helene.delatte@cirad.fr (HD)

**Data Availability Statement:** All relevant data are within the manuscript and its Supporting Information files, and All sequences were

## Abstract

Since several years, whiteflies of the species complex of *Bemisia tabaci* (Gennadius) are causing several damages on vegetable crops in Côte d'Ivoire. These sap-sucking insects are the main vector of many viruses on tomato and several species of this complex have developed resistances against insecticides. Nevertheless, there is very little information about whitefly species on vegetable crops in Côte d'Ivoire. Here, we investigated the species diversity and their genetic diversity and structuring on samples from vegetable crops in the major tomato production areas of Côte d'Ivoire. To assess this diversity, 535 whitefly samples from different localities and plant species were collected and analysed with nuclear (microsatellite) and mitochondrial (mtCOI) markers. In each site, and ecological data were recorded, including whiteflies abundance and plant species colonised by *B. tabaci*. The analysis of mtCOI sequences of whiteflies indicated the presence of four cryptic species on tomato and associated crops in Côte d'Ivoire. These were MED ASL, MED Q1, SSA 1 and SSA3. The MED ASL species dominated over all samples in the different regions and plant species. One haplotype of MED ASL out of the 15 identified predominated on most plant species and most sites. These results suggested that MED ASL is probably the main phytovirus vector in the Ivorian vegetable cropping areas. In contrast, only five haplotypes of MED Q1 were identified on vegetables but in the cotton-growing areas of the country. Its low prevalence, low nuclear and mitochondrial diversity might indicate a recent invasion of this species on vegetable crops in Côte d'Ivoire. The Bayesian nuclear analysis indicated the presence of hybrid genotypes between the two main species MED ASL and MED Q1, however in low prevalence (10%). All these results highlight the need to maintain whitefly populations monitoring for a more effective management in Côte d'Ivoire.

submitted to GenBank accession numbers
ON479242 to ON479263.

**Funding:** The role(s) played are indicated below:
HD, CS were funded by the European Union (ERDF,
contract GURDT I2016-1731-0006632), the
Conseil Régional de la Réunion and CIRAD. AJN
was funded by a PhD fellowship from the French
Ministry of Foreign Affairs via the C2D fund
'Contrat de Désendettement et de Développement,
Volet N°2 de recherche du projet AMRUGE-CI 2
(Appui à la Modernisation et à la Réforme des
Universités et Grandes Ecoles de Côte d'Ivoire)'
through the project entitled 'Adaptation et
Développement de la culture protégée dans les
conditions climatiques de la Côte d'Ivoire
(HortiNet-CI)'. LF, TM were funded by the French
Ministry of Foreign Affairs via the C2D fund
'Contrat de Désendettement et de Développement,
Volet N°2 de recherche du projet AMRUGE-CI 2
(Appui à la Modernisation et à la Réforme des
Universités et Grandes Ecoles de Côte d'Ivoire)'
through the project entitled 'Adaptation et
Développement de la culture protégée dans les
conditions climatiques de la Côte d'Ivoire
(HortiNet-CI)'. The funders had no role in study
design, data collection and analysis, decision to
publish, or preparation of the manuscript.

**Competing interests:** The authors have declared
that no competing interests exist.

# Introduction

The whitefly *Bemisia tabaci* Gennadius (Hemiptera: Aleyrodidae) is a significant pest that damages many agricultural crops worldwide. Whitefly species of this complex are extremely polyphagous insects found in field and in greenhouses in tropical and temperate regions [1, 2]. In tropical and sub-tropical countries, *B. tabaci* is a pest of primary importance, especially on crops of cassava, cotton, sweet potatoes, tobacco, and tomato [3]. In West Africa, severe population outbreaks had been observed since 1998 in cotton fields in Burkina Faso, Mali, and Côte d'Ivoire inducing severe crop damages [4]. The resulting losses had a severe impact on the economic activity of these countries, as agriculture is one of the main financial resources [3], especially in Côte d'Ivoire. *Bemisia tabaci* causes many physical damages by consuming plant sap, but are also the main vectors of many viral diseases like the *Tomato yellow leaf curl disease* (TYLCD); the *Tomato leaf curl disease* (ToLCD) on tomato or the *Cassava mosaic disease* on cassava [5, 6]. Economic losses due to *B. tabaci* are enormous. They were estimated approximately to 10 billion US dollars (USD) from 1980 to 2000 [7] in the USA and over 1 billion USD in Africa on cassava [8]. *Bemisia tabaci* is a cryptic species complex with more than 39 morphologically indistinguishable species [1, 9–16]. Most of these species are geographically localised [17] except a few, among which two are worldwide invasive: the 'Mediterranean' or 'MED' species (previously referred as 'Biotype Q') and the 'Middle East-Minor' species named 'MEAM1' (previously referred as 'Biotype B') [2]. The MED group has taxonomically recently been divided into two species: MED ASL and MED *sensus scrito*. Among these whitefly species, some of them are specialised on Solanaceae [18, 19] and can rapidly become resistant to chemical insecticides [3, 20]. As a consequence, African countries make wide use of many pesticides to control *B. tabaci* populations despite their harmful impact on potential natural enemies and on the environment. Moreover, several species of *B. tabaci* are known to carry high resistance genes so to have the best control strategies the identification of the different species are very important. Many aspects of this species complex remain unknown, such as the degree of genetic isolation between some species, the geographical distribution of the different species, or the within-species genetic diversity... Knowing of which species is present in Côte d'Ivoire is therefore extremely crucial for the management of this pest. Because these cryptic species are morphologically undifferentiable [1], we used molecular tools (mitochondrial sequences and nuclear markers) to identify the different species, and then the same molecular tools allowed us to study the genetic diversity and gene flows between populations of the different whitefly species of the *B. tabaci* species complex associated with vegetable crops in production areas in Côte d'Ivoire.

# Materials and methods

## Whiteflies sampling

This study was carried out in West Africa, actually in Côte d'Ivoire, which is bordered by the Atlantic Ocean in the south, Ghana in the east, Burkina Faso and Mali in the north, and Liberia and Guinea in the west (Fig 1). *B. tabaci* adults were collected during September to February in the main vegetable crops production areas listed by the Centre National de Recherche Agronomique (CNRA) in Côte d'Ivoire on different host plants. Sampling was carried out on 13 sites in the main production areas (Fig 1, Table 1). Those sites were chosen in the main tomato growing areas of Côte d'Ivoire where high populations of whiteflies were reported. The sampling sites were drawn using QGIS 2.8 (open maps) with shapefile data downloaded online (https://gadm.org/maps.html) and GPS coordinates of sites. Whiteflies abundance per field was determined as the total number of adults on the first five leaves of 10 randomly selected

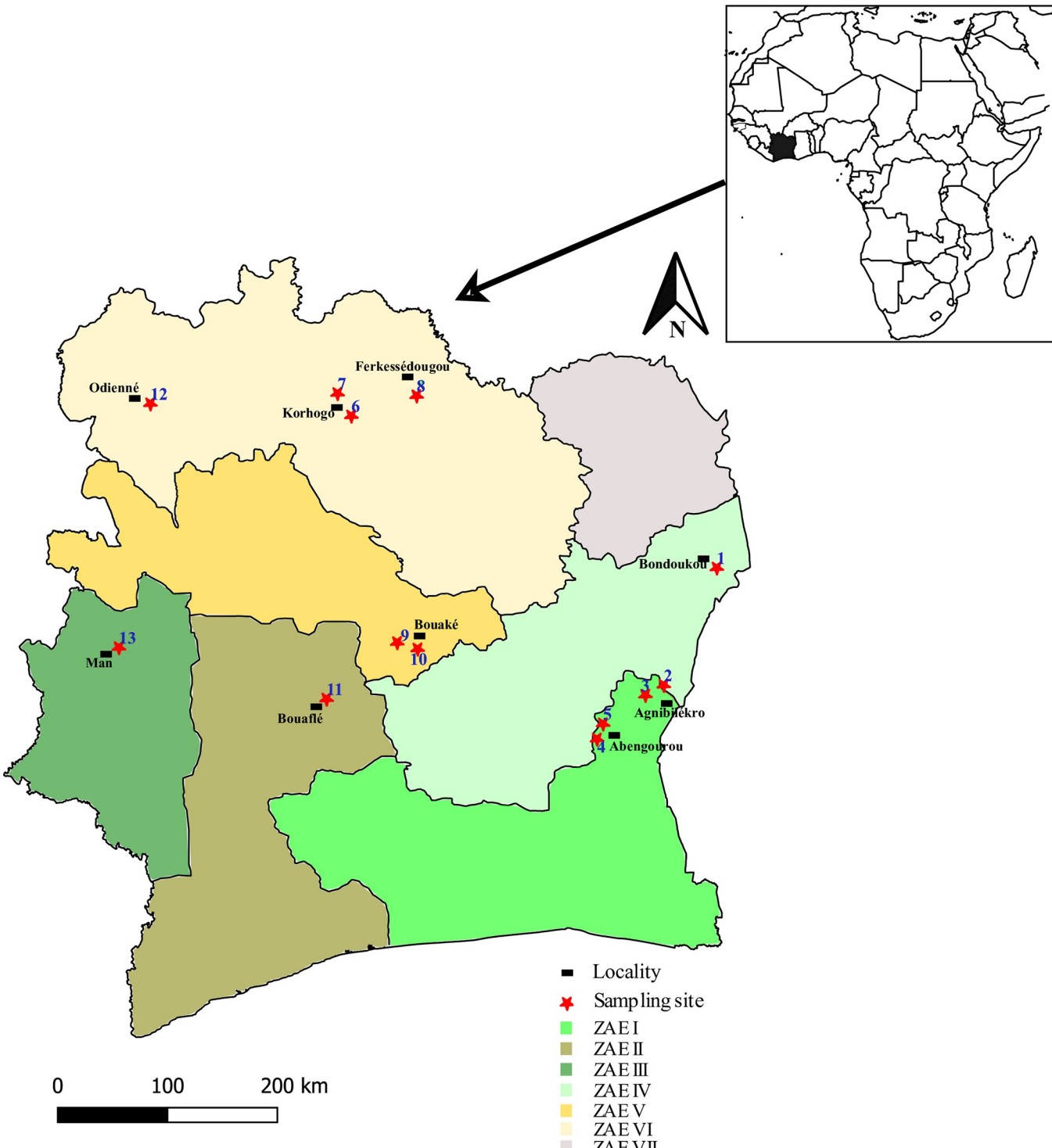

**Fig 1. Geographical distribution of *B. tabaci* collection sites in vegetable crops areas in Côte d'Ivoire.** ZAE I: southern dense humid forest zone; ZAE II: western dense humid forest zone; ZAE III: western semi-mountainous forest zone; ZAE IV: zone of dense humid semi-deciduous forest; ZAE V: forest transition zone; ZAE VI: humid tropical savanna zone; ZAE VII: dry tropical savanna zone. *1: Flatchèdougou; 2: Groupement Abadjam; 3: Miankouadiokro; 4: Kodjinan; 5: Station CNRA Abengourou; 6: Nahoualakaha; 7: Namonka; 8: Kroukrosso; 9: Béoukro; 10: Djébonoua; 11: Congoyebouesso; 12: Linguinsso; 13: Gueupleu.* Map were built using QGIS 2.8 (open maps) with shapefile data downloaded from https://gadm.org/maps.html (11/02/2022). The agro-ecological areas have been defined according to Halle and Bruzon [22].

**Table 1. Numbers of *B. tabaci* genetic groups according to localities and plant species.**

| Locality | Site | GPS Coordinates | | Date | Plants species | | | NT | NW ± ES | N. of *B. tabaci* species | | | |
|---|---|---|---|---|---|---|---|---|---|---|---|---|---|
| | | Latitude | Longitude | | Scientific name | Common name | Age (months) | | | MED ASL | MED Q1 | SSA 1 | SSA 3 |
| **Bondoukou** | 1 | 8.1969444 | -2.7680555 | Janv-19 | *Solanum melongena* L. | Eggplant | 3 | 4 | 120±44 | 29 | 0 | 0 | 0 |
| | | | | | *Solanum lycopersicum* L. | Tomato | 2 | 4 | 43±14 | 30 | 0 | 0 | 0 |
| **Agnibilékro** | 2 | 7.2380556 | -3.2016666 | Janv-19 | *Solanum lycopersicum* L. | Tomato | 1.5 | 4 | 460±40 | 32 | 0 | 0 | 0 |
| | | | | | *Brassica oleracea* L. | Cabbage | 1 | 4 | 1±0.1 | 32 | 0 | 0 | 0 |
| | 3 | 7.1566667 | -3.3519444 | Janv-19 | *Solanum lycopersicum* L. | Tomato | 2 | 4 | 290±70 | 31 | 0 | 0 | 0 |
| **Abengourou** | 4 | 6.7980556 | -3.7477777 | Janv-19 | *Solanum lycopersicum* L. | Tomato | 1 | 2 | 201±66 | 31 | 0 | 0 | 0 |
| | 5 | 6.9225000 | -3.6983333 | Janv-19 | *Solanum melongena* L. | Eggplant | 6 | 4 | 10±5 | 29 | 0 | 0 | 0 |
| **Korhogo** | 6 | 9.4394444 | -5.7577777 | Janv-19 | *Solanum lycopersicum* L. | Tomato | 2 | 4 | 3±1 | 21 | 1 | 0 | 0 |
| | | | | | *Solanum melongena* L. | Eggplant | 3 | 4 | 16±6 | 28 | 4 | 0 | 0 |
| | 7 | 9.6227778 | -5.8702777 | Janv-19 | *Solanum lycopersicum* L. | Tomato | 1 | 2 | 30±7 | 17 | 1 | 0 | 0 |
| **Ferkessédougou** | 8 | 9.6041667 | -5.2233333 | Janv-19 | *Solanum lycopersicum* L. | Tomato | 2 | 3 | 14±6 | 16 | 16 | 0 | 0 |
| | | | | | *Cucumis sativus* L. | Cucumber | 3 | 5 | 26±10 | 12 | 19 | 0 | 0 |
| **Bouaké** | 9 | 7.5655556 | -5.3188888 | Janv-19 | *Solanum lycopersicum* L. | Tomato | 1 | 7 | 30±10 | 29 | 2 | 0 | 0 |
| | 10 | 7.5341667 | -5.2783333 | Janv-19 | *Solanum lycopersicum* L. | Tomato | 3 | 6 | 1±0.1 | 26 | 3 | 2 | 1 |
| **Bouaflé** | 11 | 7.1255556 | -5.96 | Janv-19 | *Solanum lycopersicum* L. | Tomato | 2 | 8 | 38±10 | 29 | 0 | 0 | 0 |
| **Odienné** | 12 | 9.5387031 | -7.4011030 | Fev-19 | *Solanum lycopersicum* L. | Tomato | 3 | 5 | 255±67 | 31 | 0 | 0 | 0 |
| **Man** | 13 | 7.5461111 | -7.6588888 | Fev-19 | *Abelmoschus esculentus* L. | Okra | 4 | NA | 340±65 | 32 | 0 | 0 | 0 |
| | | | | | *Phaseolus vulgaris* L. | Green bean | 1 | 1 | 170±55 | 31 | 0 | 0 | 0 |
| **Total N. of *B. tabaci* species** | | | | | | | | | | 486 | 46 | 2 | 1 |

NT: number of days elapsed between the last whitefly treatments and sample collection, NW: average number of whiteflies observed per plant (noted on the 5 upper leaves).

plants on the diagonals of field sampled (rating scale: 1–9 whiteflies = 1; 10–49 whiteflies = 10; 50–99 whiteflies = 50; 100–499 whiteflies = 100 and over 500 whiteflies = 500). A minimum of 100 individuals was sampled in each site from each host plant species in a random diagonal pattern to avoid collecting related individuals as much as possible. The whiteflies were collected on african eggplant (*Solanum ethiopicum*), cabbage (*Brassica oleracea*), tomato (*Solanum lycopersicum*), okra (*Abelmoschus esculentus*), cucumber (*Cucumus sativus*) and green bean (*Phaseolus vulgaris*). After sampling, whiteflies were stored in 95% ethanol at -20˚C and morphologically sexed under binocular loupe [21], before DNA extraction. In each sampled field a very simple survey was carried out, asking the plant variety, the age of the crop, if whitefly treatments were applied, and the date of the last insecticide treatment made.

## DNA extraction

Only female individuals of *B. tabaci* were used in this study due to their diploidy. The protocol of Delatte and colleagues [18] was used for DNA extraction. All collected females with a maximum of 31 per plant species and sites were incubated in 50 μl of extraction buffer [50 mmol/l KCl, 10 mmol/l Tris-base (pH 8), 0.45% IGEPAL 630, 0.45% Tween 20 and 500 mg/ml of proteinase K (Sigma)] at 65˚C for 20 h, then a final incubation step was performed at 95˚C for 10 min to inhibit the proteinase K. The extracts were finally stored at -20˚C after a brief centrifugation, until use.

## Mitochondrial DNA amplification and sequences analysis

Polymerase chain reaction (PCR) reactions were performed as described by Ally and colleagues [23], using the primer pair designed by Mugerwa [24] (S1 Table). The PCR reaction mix was

prepared with a final volume of 25 μl, containing 12.5 μl of 2x type-it master mix (This Master Mix contains a specific Taq DNA Polymerase adapted to SSR types of markers; Qiagen; France); 8 μl of HPLC water; 1.25 μl of each primer and 2 μl of DNA. The PCR programme was as follow: an initial denaturation step at 95˚C for 15 minutes; followed by 40 cycles of denaturation at 95˚C for 30 seconds; hybridisation at 52˚C for 30 seconds; extension at 72˚C for one minute and then a final extension step at 72˚C for 10 minutes. The amplified fragments (PCR products) were analyzed with an automated DNA/RNA analyser QIAxcel Advanced (QIAxel ScreenGel® software; Qiagen). The PCR amplicons were then sent to the Macrogen Europe laboratory for sequencing. The sequences produced were then compared to existing sequences using the BLAST algorithm in GenBank (http://www.ncbi.nm.nih.gov) and aligned with reference sequences of the *B. tabaci* species complex from the literature. The sequences were manually edited and aligned using Geneious prime software version 2019.1.3. DnaSP v.6 [25] was used to obtain the number and distribution of haplotypes in our sequences. The selected sequences and reference sequences from the literature were aligned using ClustalW [26] before being submitted to Jmodeltest 2.1.10 [27] to determine the best nucleotide substitution model. The phylogenetic tree was created using MrBayes with 1 million of iterations and 4 heated chains [28].

## Microsatellite genotyping

A total of 535 adult females was genotyped using 12 microsatellites loci (S1 Table). PCR was performed as described by Delatte and colleagues [18]. The equivalent of 10 ng of genomic DNA was used for amplification in 96-well plates with the QIAGEN multiplex PCR Master Mix kit according to the manufacturer's instructions, in 15 μL. Three mixes of primer were used: MS145, P59, P7, WF2H06 (mix 1); WF1G03, WF1D04, P5 (mix 2) and CIRSSA2, CIRSSA6, CIRSSA7, CIRSSA13, CIRSSA41 (mix 3). The PCR programmes were as follow, 15 min at 95˚C followed by 40 cycles of: 30 s at 95˚C; 90 s at 55˚C; 1 min at 72˚C; with a final extension step of 60˚C for 15 min (mix 1 and mix 2) and 15 min at 95˚C followed by 40 cycles of 30 s at 95˚C, 90 s at 56˚C; 1 min at 72˚C; with a final extension step of 60˚C for 15 min (mix 3). 2 μl of final PCR products were mixed with a mixture of 10.7 μl of Hi-Di-formamideTM (Applied Biosystems) and 0.3 μl of size marker (GeneScan 500Liz) and denatured at 95˚C for 5 min. Fragments were analysed on an ABI Prism 3100 Automated Genetic Analyzer (Applied Biosystems) and genotypic data were viewed and analysed manually using Gene-Mapper software version 4.0.

## Analysis of genetic and structural diversity

To analyse our data the Bayesian classification program STRUCTURE version 2.3.3 was used [29]. This program differentiates genetic clusters according to the allele frequency at each locus and assigns posterior probabilities of assignment to a given cluster for each individual. To determine the optimal number of clusters or genetically different populations (K) in our sample, STRUCTURE was run for K values between 1 and 10 (each K was run 15 times). The *ad hoc* method of Evanno [30] was applied to find the optimal number of K in the dataset using structure Harvester (http://taylor0.biology.ucla.edu/structureHarvester/) and Clumpak (http://clumpak.tau.ac.il/) online.

To use STRUCTURE, it is assumed that populations are in Hardy-Weinberg equilibrium (HWE) and the association between alleles within loci is random, i.e., no linkage disequilibrium exists. Both hypotheses were subsequently tested on each cluster using the exact tests implemented in the Genepop 4.0 software [31]. Allelic frequencies, observed (Ho) and expected (He) heterozygosity, and excess of heterozygotes (FIS) were analysed with Genepop 4.0 and Genetix 4.01 software. The pairwise FST genetic distance matrix between populations of each species was obtained using Genepop 4.0 software [32]. Isolation by distance (IBD) was

tested using the Mantel test between the genetic distance matrices and the geographical distance matrix generated from GPS coordinates with R software, using Genepop 4.0 software.

## Results

### Distribution and prevalence of cryptic species of *B. tabaci*

The average whiteflies per plant species ranged from 1 (site 2 on cabbage and site 9 on tomato) to 460 (site 2 on tomato). As a highlight, a relatively high number of whiteflies was observed on green bean (about 170 whiteflies/plant) despite the fact that the whiteflies treatments were carried out 1 day before the sampling.

Sequencing of the partial cytochrome oxidase I (mtCOI) of whitefly females revealed four whitefly species on different plants in the nine localities (13 sites) surveyed (Table 1): MED ASL, MED Q1, SSA1 and SSA3. The data set includes 90.84% of MED ASL (n = 486), 8.60% of MED Q1 (n = 46), 0.37% of SSA1 (n = 2) and 0.19% of SSA3 (n = 1). In all surveyed localities, MED ASL was the most observed, with a predominance in most of the sites (11 sites) located in the forest areas of the country. MED ASL was found on five different crop families (Solanaceae, Malvaceae, Brassicaceae, Fabaceae, Cucurbitaceae). MED Q1 was found in sympatry with MED ASL, only in three localities (5 sites) located in the North (Korhogo and Ferkessédougou) and the centre (Bouaké), with a higher prevalence in the North on two different plant families (Solanaceae and Cucurbitaceae). Moreover, SSA1 and SSA3 were detected in the same site (Bouaké) in sympatry with MED ASL and MED Q1. However, SSA1 and SSA3 were excluded from further analysis due to their very low numbers.

### Phylogenetic analysis

Analysis of mitochondrial sequences obtained after sequencing showed genetic diversity within *B. tabaci* species from one area to another. *Bemisia tabaci* mtCOI sequences for all haplotypes identified in this study were submitted to GenBanK accession numbers: ON479242 to ON479263. Fifteen (15/486) haplotypes were identified for MED ASL and five (5/46) for MED Q1. The dominant haplotype (P2A6_CI_2019_ON479261) for MED ASL was found on all plant species surveyed while the dominant haplotype (P5G2_CI_2019_ON479244) for MED Q1 was found only on tomato, eggplant and cucumber (Fig 2). These haplotypes shared the highest nucleotide identities with the accession MH205754.1 (100% nucleotide identity) detected in Uganda on okra for MED ASL and MH205752.1 (99.85% nucleotide identity) for MED Q1 detected in Sudan on *Cucurbita* sp. [33]. A single haplotype of SSA1 (P5E5_CI_2019_ON479262; 100% nucleotide identity with MK532685.1; Nigeria; Direct Submission) and SSA3 (P5H7_CI_2019_ON479263; 99.53% nucleotide identity with KM377923; Nigeria) [34] were found (Fig 2; S2 Table).

### Genetic diversity and population structure

A total of 526 whiteflies were successfully genotyped with the 12 microsatellite primer pairs. The mean number of alleles at all loci per population ranged from 5.5 (site 7) to 7.91 (site 13) for MED ASL and from 2.17 (site 6) to 5.58 (site 8) for MED Q1 (Table 2). The mean allelic richness was higher for MED ASL (4.6 ± 0.06) than for MED Q1 (1.34 ± 0.06). For all populations surveyed, Ho was lower than He with *Fis* values ranging from 0.14 to 0.33 for MED ASL and from 0.33 to 0.51 for MED Q1. All populations showed a significant deviation from HWE for MED ASL and MED Q1 (Table 2). Within these species, genetic differentiation between populations in two sites was weak (*Fst* ranging from -0.009 to 0.0681), but significant in 94%

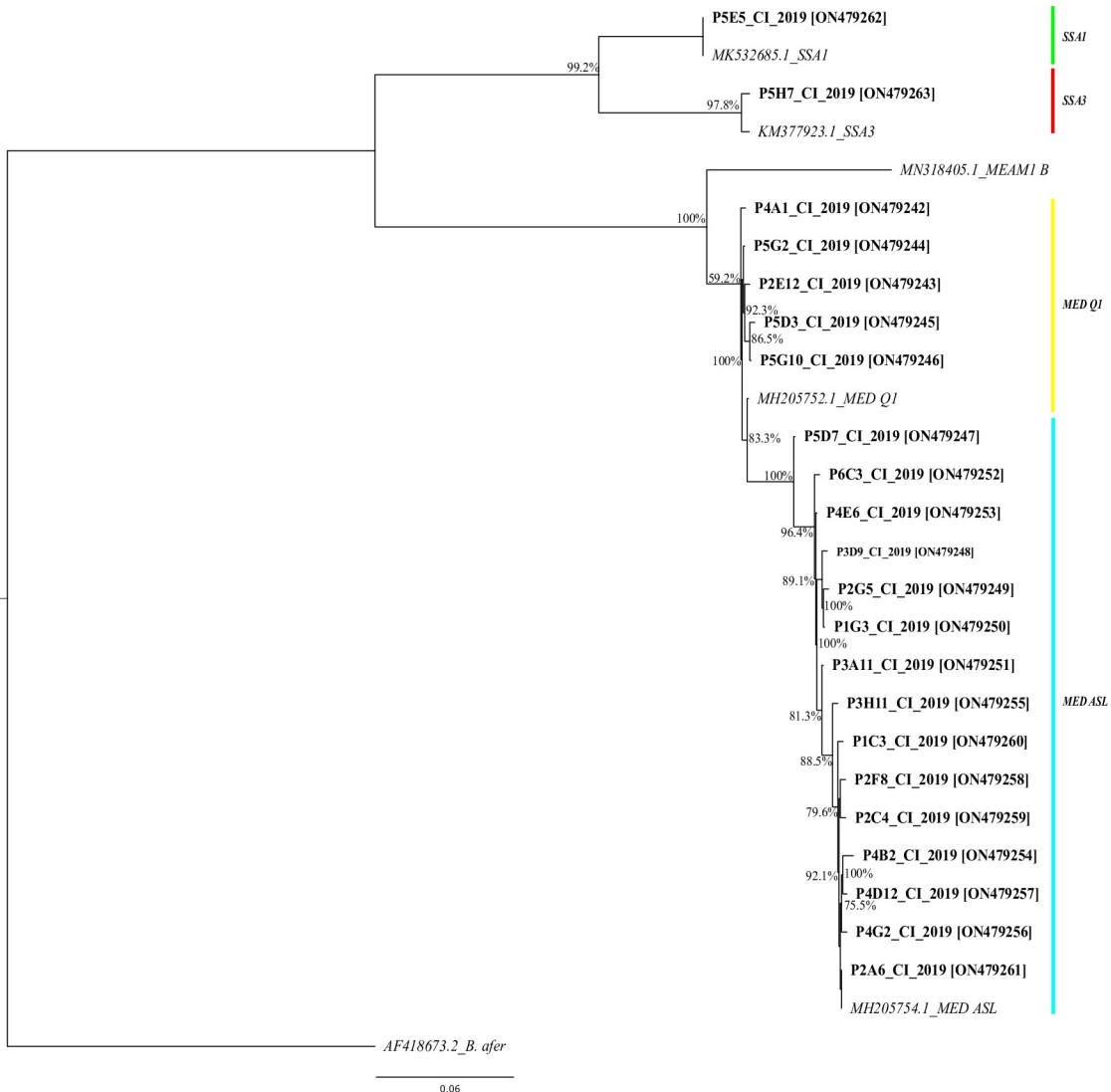

**Fig 2. Phylogenetic tree showing the relationships between *B. tabaci* mitochondrial COI DNA sequences for all 22 haplotypes identified in this study (S2 Table) and five representative samples of nearby sequences available on NCBI.** *B. afer* (AF418673.2) was used as outgroup. The tree was built using MrBayes. The numbers associated with the nodes indicate the posterior probability for those nodes. The names of the haplotypes sequences characterized in this study and their accession number are in bold.

of cases (73 pairs/78; S3 Table) for MED ASL and not significant for MED Q1 (only two populations were considered with numbers of more than 5 individuals: *Fst* = -0.0126).

Bayesian analyses revealed a weak sub-structuration within the MED ASL species (Fig 3A and 3B), with the best number of clusters which was 3 (method of Evanno) [30], and no sub-structure between populations of MED Q1 (Fig 3A). The three clusters detected within MED ASL did not reflect any differentiation between sampled sites, between collected plant species or agroecological zones. Furthermore no significant correlation between genetic and geographical distances of the populations of these two species were detected (Mantel Test, P>0.05).

These Bayesian analyses revealed also the presence of interspecific hybrid individuals (MED ASL-MED Q1). Within the MED ASL individuals (identified using their mitochondrial sequence) a few individuals (n = 53/486) were considered as hybrids when their Bayesian posterior

**Table 2. Genetic diversity indices according to the genetic groups of *B. tabaci* and the sites surveyed.**

| Species | Sites | *N* | *Na* | *Ra* | *Ho* | *He* | *Fis (W&C)* |
|---|---|---|---|---|---|---|---|
| **MED ASL** | 1 | 59 | 7.08 | 4.45 | 0.27 | 0.41 | 0.33[***] |
| | 2 | 64 | 7.75 | 4.57 | 0.29 | 0.43 | 0.31[***] |
| | 3 | 31 | 6.08 | 4.59 | 0.33 | 0.47 | 0.30[***] |
| | 4 | 31 | 6.17 | 4.38 | 0.34 | 0.45 | 0.25[***] |
| | 5 | 29 | 6.17 | 4.65 | 0.38 | 0.43 | 0.14[***] |
| | 6 | 49 | 7.33 | 4.73 | 0.34 | 0.42 | 0.20[***] |
| | 7 | 17 | 5.50 | 4.84 | 0.32 | 0.41 | 0.25[***] |
| | 8 | 28 | 6.58 | 5.05 | 0.36 | 0.45 | 0.21[***] |
| | 9 | 29 | 5.91 | 4.47 | 0.34 | 0.42 | 0.20[***] |
| | 10 | 26 | 6.00 | 4.71 | 0.32 | 0.42 | 0.23[***] |
| | 11 | 29 | 5.75 | 4.17 | 0.31 | 0.39 | 0.20[***] |
| | 12 | 31 | 5.91 | 4.55 | 0.32 | 0.44 | 0.27[***] |
| | 13 | 63 | 7.91 | 4.63 | 0.30 | 0.44 | 0.29[***] |
| **MED Q1** | 6 | 5 | 2.17 | 1.28 | 0.13 | 0.25 | 0.51[***] |
| | 8 | 35 | 5.58 | 1.40 | 0.26 | 0.39 | 0.33[***] |

*N*: number of individuals collected, *Na*: mean number of alleles/locus, *Ra*: allelic richness, *Ho*: observed heterozygosity, *He*: expected heterozygosity, *Fis*: Fixation indices. *P*-value from Hardy-Weinberg equilibrium test, are indicated (*, $P < 0.05$; **, $P < 0.01$; ***, $P < 0.001$).

probability of individual assignment to the MED Q1 genetic cluster was over 30%. The same screening was performed for the MED Q1 individuals, and within these individuals 3/46 were found with an assignment to the MED ASL genetic cluster. The mtCOI haplotype (maternally transmitted) gives the direction of the initial cross for hybrids: a MED ASL mtCOI haplotype indicates a cross between a MED ASL female and a MED Q1 male (and vice versa) [35]. Consequently, we identified slightly more interspecific matings with MED ASL females than with MED Q1 females (10% versus 6%). The distribution of the assignment probabilities within MED ASL indicates that most of the hybrids derived from backcrosses, mostly toward the MED Q1 species (1.85% (n = 9) have posterior probabilities of individual assignment above 80% to MED Q1) and are rather located in the north towards the cotton areas where both species are found in sympatry.

Numbers below the x-axis correspond to the sampling sites. Black lines separate individuals between each site. MED ASL was the only species for which a substructure (K = 3) was observed with Bayesian analysis.

## Discussion

Spectacular outbreaks of *B. tabaci* populations have been observed, contrasting with the abundance values recorded from 2014 to 2015 by Didi and colleagues (17.29 adult whiteflies /30 plants) [36] in Côte d'Ivoire on the cotton crop during periods of low rainfall. According to Legg [37], super-abundant populations on crops of *B. tabaci*, usually have more than 100 adults on the upper five leaves of the plant. According to this definition, Côte d'Ivoire belongs to the category of super-abundant whitefly countries at the time of our study. Several factors could explain this abundance of *B. tabaci* on vegetable crops in Côte d'Ivoire, including climate change and plant characteristics [36]. Regarding climate change, the unavailability of precise data on the different regions limits our possibilities to go further testing this hypothesis. However, it has been shown that rainfall significantly limits the development of *B. tabaci* populations [38]. However, a significant decrease in rainfall has been reported in recent years in Côte

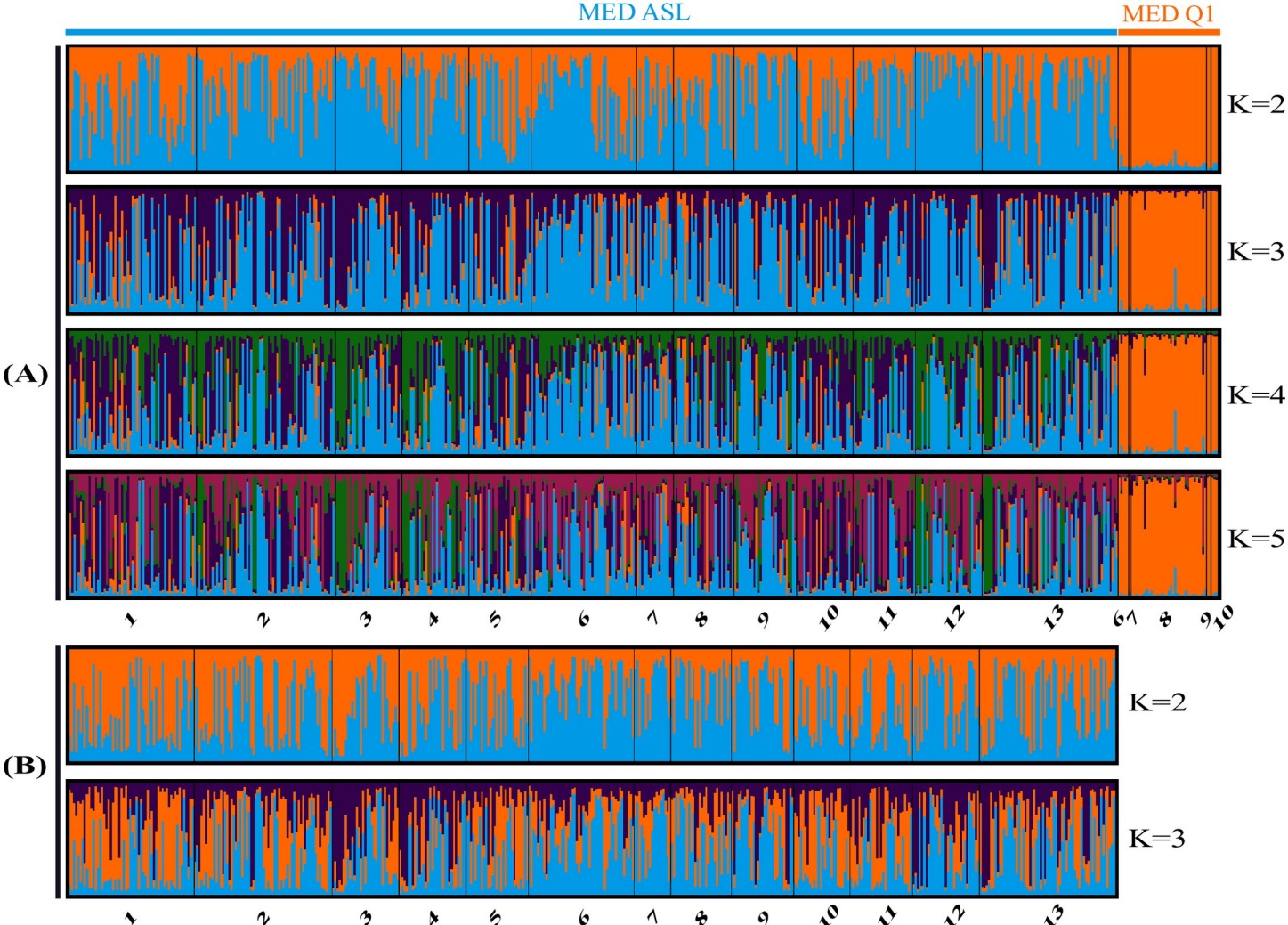

**Fig 3.** Bar plot representing population structures of *B. tabaci* from Côte d'Ivoire organised by population and site at K = 2; 3; 4 and 5: (A) MED ASL and MED Q1 (n = 532); (B) Only MED ASL (n = 486) analysed with 12 microsatellite markers.

d'Ivoire [22, 39]. Thus, the decrease in heavy rainfall in recent years could be, in this context, one of the plausible causes of the proliferation of whitefly populations.

This first analysis of cryptic *B. tabaci* species allowed us to identify the major species present in the main vegetable crops production areas in Côte d'Ivoire. Thus, four species of *B. tabaci* were identified on tomato, but also on other vegetable crops in Côte d'Ivoire, especially MED ASL; MED Q1; SSA 1 and SSA3. The prevalence of these species were different according to the sites sampled. The MED Q1 species was present mostly in the north, in the cotton zone, but in low proportions in the centre of the country. One could hypothesized that the low abundance per site, short distribution, and low genetic diversity observed could be the result of a recent introduction of this species into the country (via the northern borders) and cultivation practices that might have facilitated its spread, especially regarding to pesticide use. Indeed, this species is recognized to be invasive worldwide, a very good competitor and carry in some populations insecticide resistance genes [3, 40, 41]. According to Doumbia and Kwadjo [42], the majority of market garden farmers do not respect the recommended dosage of pesticides used. However, most insecticides used on cotton are diverted to vegetable crops [43]. A recent

study conducted by Didi and colleagues [44] showed a predominance of MED Q1 on cotton which is mainly grown in the north and centre of Côte d'Ivoire. Therefore, what was observed here could indeed be a migration of MED Q1 from this crop to vegetable crops (Malvaceae; Solanaceae; Fabaceae etc.). Whiteflies, which are extremely polyphagous and migrate between crops, could therefore find in constant presence of the same active ingredients and this could in the long-term select resistant populations. Horowitz and colleagues [41, 45] showed in Israel that invasive populations of MED Q1 species could rapidly develop resistance to neonicotinoids after more than 20 generations of maintenance under controlled conditions, highlighting the significant capacity of this species to rapidly acquire this type of resistance to this family of chemicals, but also to some chemicals of the Organophosphate family. This property would make MED Q highly competitive when sharing its ecological niche with other cryptic *B. tabaci* species and could maybe replace the resident population with increased numbers and highly insecticide resistant populations. These populations should therefore be tested for their resistance to the most commonly used insecticide active ingredients in order to find better management methods (control) and prone more respectful control measures for preserving the farmers' health and the environment.

The MED ASL species was detected on all of the sites and plants surveyed and in large proportions, except in the north, where it was less frequent and always in sympatry with MED Q1. This geographic distribution observed could be linked to several hypotheses, i) our sampling targeting only vegetable crops could generate an interpretation bias, ii) the northern zone would be less favourable to the development of this species than the more humid southern zones, iii) a competition phenomenon between MED Q and MED ASL species on the same resources, with MED Q prevailing over MED ASL, iv) the populations of MED ASL species in these zones would be more sensitive to insecticide treatments used. The sampling carried out for this study was only done on vegetable crops, but in a similar way in all of the sites and regions surveyed, allowing an idea of the diversity of species present on these crops. A comparative study of the resistance levels of the MED ASL and MED Q populations in these different agroecological zones would make it possible to confront some of these hypotheses. Indeed, it has been shown that the *B. tabaci* IO species, indigenous of the Indian ocean islands, was highly susceptible to insecticide compared to the invasive MEAM1 species and allowed MEAM1 to be dominant in crops, particularly in areas where phytosanitary pressure is high [35, 46]. Similarly, comparative studies of life history traits of populations of MED ASL and MED Q1 in Côte d'Ivoire could also provide a better understanding of their distributions and the interactions between these species. MED ASL has already been reported in sympatry with MED Q1 on cotton and vegetable crops with a high proportion of MED Q1 in Burkina Faso and in the cotton areas of Côte d'Ivoire by several studies [44, 47–49], which is the opposite of our results. This could be explained by different phytosanitary practices and/or climatic and environmental conditions and would support our hypotheses listed above. In view of the predominance of MED ASL on most sites sampled in the major vegetable crops production areas, it might be suggested that this is the group mainly responsible for the spread of vegetable crops viruses in Côte d'Ivoire and the increased incidence of the disease that they cause [50–52].

Molecular analyses of the samples (nuclear and mitochondrial) were able to clearly separate the MED ASL and MED Q1 populations into two distinct groups. The difference between these two cryptic species was also highlighted by Vyskočilová and colleagues [33] showing not only genetic differentiation but also reproductive isolation in the laboratory between MED ASL and MED Q populations. Nevertheless, in the analysed samples of our study, we observed a very low number (10%) of individuals with potential traces of hybridisation between the two species, the two main species present on vegetable crops in Côte d'Ivoire, in our dataset. This confirms that reproductive isolation between the two species is incomplete. Slightly more

hybrids where found with individuals bearing a MED ASL mtCOI, which may be explained by the predominance of this species in the sampled sites. This phenomenon has already been observed in cryptic species and even between other species of the *B. tabaci* species complex (i.e. IO and MEAM1) [53, 54]. These very restricted gene flows are generally very uncommon and reproductive isolation is maintained between species [55]. This possible gene flow could be a pathway for the passage, for example of insecticide resistance genes between species, especially with MED Q1, which can rapidly become resistant to insecticide [56]. However, further analyses are needed, in particular by testing the resistance levels of these two cryptic species.

The nuclear molecular analysis conducted on MED ASL populations revealed structuration of the populations into three genetic clusters, which showed no relationship to plants and geographical areas, and no evidence of genetic isolation by distance between sites was detected. This sub structuration is found at the level of each site and each of the clusters are found in all sampled sites. These results indicate that i) exchanges of individuals between fields and even different agroecological zones of Côte d'Ivoire are frequent, facilitated by the movement of horticultural crops, vegetable production sites and exchanges of plant material, ii) there may be reproductive isolation between some individuals at site level that may produce these deviations from the Hardy-Weinberg equilibrium and this low level of sub structuration observed.

The unusual detection of SSA1 and SSA3 species, which are often more closely related to cassava, on tomato could be explained by a potentially transient movement of some individuals from their favourite plant species (cassava) to tomato. Indeed, the site where these two species were identified was bounded on one side by cassava crops and on the other side by a cassava fallow. A previous study conducted in sub-Saharan Africa [57] had already highlighted the fact that whitefly populations developing on cassava appeared to be limited to cassava. However, our study reveals that these species, although preferentially on cassava, can also be found on other plants, including tomato. Recent studies describing whiteflies on cassava in East Africa have found similar results [23, 58].

All these results highlight the need to maintain whitefly populations monitoring for a more effective management and to look for other alternative strategies for pest management in West Africa for a more sustainable agriculture.

## Supporting information

**S1 Table. Loci used for nuclear and mitochondrial DNA analysis.**
(DOCX)

**S2 Table. All *Bemisia tabaci* mtCOI haplotypes and their accession numbers found in this study.**
(DOCX)

**S3 Table. FST matrix of MED ASL populations.** All bolded numbers are significant (Bonferroni corrected p-value).
(DOCX)

## Acknowledgments

The authors would like to thank all the farmers of Côte d'Ivoire where the sampling was performed and the CNRA for facilitating the sampling. The authors acknowledge the Plant Protection Platform (CIRAD 3P) where all molecular experiments were conducted.

## Author Contributions

**Conceptualization:** Anthelme-Jocelin N'cho, Koutoua Seka, Thibaud Martin, Hélène Delatte.

**Data curation:** Anthelme-Jocelin N'cho, Christophe Simiand, Hélène Delatte.

**Formal analysis:** Anthelme-Jocelin N'cho, Christophe Simiand, Hélène Delatte.

**Funding acquisition:** Lassina Fondio, Thibaud Martin, Hélène Delatte.

**Investigation:** Anthelme-Jocelin N'cho, Kouamé Patrice Assiri, Daniel H. Otron, Kouassi Arthur Jocelin Konan, Marie-France Kouadio.

**Methodology:** Anthelme-Jocelin N'cho, Hélène Delatte.

**Project administration:** Lassina Fondio, Thibaud Martin.

**Resources:** Germain Ochou, Hortense Atta Diallo, Hélène Delatte.

**Supervision:** Koutoua Seka, Thibaud Martin, Hélène Delatte.

**Validation:** Koutoua Seka, Hortense Atta Diallo, Thibaud Martin, Hélène Delatte.

**Visualization:** Anthelme-Jocelin N'cho, Hélène Delatte.

**Writing – original draft:** Anthelme-Jocelin N'cho.

**Writing – review & editing:** Anthelme-Jocelin N'cho, Koutoua Seka, Thibaud Martin, Hélène Delatte.

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
