## [Decision Letter · Decision Letter 0]

4 Jul 2022

PONE-D-22-15957Genetic diversity of whitefly species of the Bemisia tabaci (Gennadius) (Hemiptera: Aleyrodidae) species complex, associated with vegetable crops in Côte d'Ivoire.PLOS ONE

Dear Dr. N'cho,

Thank you for submitting your manuscript to PLOS ONE. After careful consideration, we feel that it has merit but does not fully meet PLOS ONE’s publication criteria as it currently stands. Therefore, we invite you to submit a revised version of the manuscript that addresses the points raised during the review process.

We look forward to receiving your revised manuscript.

Kind regards,

Yunzhou Li

Academic Editor

PLOS ONE

Journal Requirements:

When submitting your revision, we need you to address these additional requirements. 1. Please ensure that your manuscript meets PLOS ONE's style requirements, including those for file naming. The PLOS ONE style templates can be found at https://journals.plos.org/plosone/s/file?id=wjVg/PLOSOne_formatting_sample_main_body.pdf and https://journals.plos.org/plosone/s/file?id=ba62/PLOSOne_formatting_sample_title_authors_affiliations.pdf
 2. We note that the grant information you provided in the ‘Funding Information’ and ‘Financial Disclosure’ sections do not match.  When you resubmit, please ensure that you provide the correct grant numbers for the awards you received for your study in the ‘Funding Information’ section. 3. Thank you for stating the following in the Acknowledgments Section of your manuscript:  "This study was funded by the CIRAD, the ‘Conseil Régional de La Réunion’ and the European Agricultural Fund for Rural Development (EAFRD)." We note that you have provided funding information that is not currently declared in your Funding Statement. However, funding information should not appear in the Acknowledgments section or other areas of your manuscript. We will only publish funding information present in the Funding Statement section of the online submission form. Please remove any funding-related text from the manuscript and let us know how you would like to update your Funding Statement. Currently, your Funding Statement reads as follows:  "The role(s) played are indicated below:HD, CS  were funded by the European Union (ERDF, contract GURDT I2016-1731-0006632), the Conseil Régional de la Réunion and CIRAD.AJN was funded by a PhD fellowship from the French Ministry of Foreign Affairs via the C2D fund ‘Contrat de Désendettement et de Développement, Volet N°2 de recherche du projet AMRUGE-CI 2 (Appui à la Modernisation et à la Réforme des Universités et Grandes Ecoles de Côte d’Ivoire)’ through the project entitled ‘Adaptation et Développement de la culture protégée dans les conditions climatiques de la Côte d’Ivoire (HortiNet-CI)’.LF, TM were funded by the French Ministry of Foreign Affairs via the C2D fund ‘Contrat de Désendettement et de Développement, Volet N°2 de recherche du projet AMRUGE-CI 2 (Appui à la Modernisation et à la Réforme des Universités et Grandes Ecoles de Côte d’Ivoire)’ through the project entitled ‘Adaptation et Développement de la culture protégée dans les conditions climatiques de la Côte d’Ivoire (HortiNet-CI)’. The funders had no role in study design, data collection and analysis, decision to publish, or preparation of the manuscript" Please include your amended statements within your cover letter; we will change the online submission form on your behalf. 4. We note that Figure 1 in your submission contain [map/satellite] images which may be copyrighted. All PLOS content is published under the Creative Commons Attribution License (CC BY 4.0), which means that the manuscript, images, and Supporting Information files will be freely available online, and any third party is permitted to access, download, copy, distribute, and use these materials in any way, even commercially, with proper attribution. For these reasons, we cannot publish previously copyrighted maps or satellite images created using proprietary data, such as Google software (Google Maps, Street View, and Earth). For more information, see our copyright guidelines: http://journals.plos.org/plosone/s/licenses-and-copyright. We require you to either (1) present written permission from the copyright holder to publish these figures specifically under the CC BY 4.0 license, or (2) remove the figures from your submission: a. You may seek permission from the original copyright holder of Figure 1 to publish the content specifically under the CC BY 4.0 license.   We recommend that you contact the original copyright holder with the Content Permission Form (http://journals.plos.org/plosone/s/file?id=7c09/content-permission-form.pdf) and the following text:“I request permission for the open-access journal PLOS ONE to publish XXX under the Creative Commons Attribution License (CCAL) CC BY 4.0 (http://creativecommons.org/licenses/by/4.0/). Please be aware that this license allows unrestricted use and distribution, even commercially, by third parties. Please reply and provide explicit written permission to publish XXX under a CC BY license and complete the attached form.” Please upload the completed Content Permission Form or other proof of granted permissions as an ""Other"" file with your submission. In the figure caption of the copyrighted figure, please include the following text: “Reprinted from [ref] under a CC BY license, with permission from [name of publisher], original copyright [original copyright year].” b. If you are unable to obtain permission from the original copyright holder to publish these figures under the CC BY 4.0 license or if the copyright holder’s requirements are incompatible with the CC BY 4.0 license, please either i) remove the figure or ii) supply a replacement figure that complies with the CC BY 4.0 license. Please check copyright information on all replacement figures and update the figure caption with source information. If applicable, please specify in the figure caption text when a figure is similar but not identical to the original image and is therefore for illustrative purposes only.The following resources for replacing copyrighted map figures may be helpful: USGS National Map Viewer (public domain): http://viewer.nationalmap.gov/viewer/The Gateway to Astronaut Photography of Earth (public domain): http://eol.jsc.nasa.gov/sseop/clickmap/Maps at the CIA (public domain): https://www.cia.gov/library/publications/the-world-factbook/index.html and https://www.cia.gov/library/publications/cia-maps-publications/index.htmlNASA Earth Observatory (public domain): http://earthobservatory.nasa.gov/Landsat: http://landsat.visibleearth.nasa.gov/USGS EROS (Earth Resources Observatory and Science (EROS) Center) (public domain): http://eros.usgs.gov/#Natural Earth (public domain): http://www.naturalearthdata.com/

Reviewers' comments:

Reviewer's Responses to Questions

**Comments to the Author**

1. Is the manuscript technically sound, and do the data support the conclusions?

Reviewer #1: No

Reviewer #2: Yes

Reviewer #3: Yes

Reviewer #4: Partly

2. Has the statistical analysis been performed appropriately and rigorously? 

Reviewer #1: No

Reviewer #2: Yes

Reviewer #3: Yes

Reviewer #4: No

3. Have the authors made all data underlying the findings in their manuscript fully available?

Reviewer #1: Yes

Reviewer #2: Yes

Reviewer #3: Yes

Reviewer #4: Yes

4. Is the manuscript presented in an intelligible fashion and written in standard English?

Reviewer #1: Yes

Reviewer #2: No

Reviewer #3: Yes

Reviewer #4: No

5. Review Comments to the Author

Reviewer #1: Genetic diversity of whitefly species of the Bemisia tabaci (Gennadius) (Hemiptera: Aleyrodidae) species complex, associated with vegetable crops in Côte d'Ivoire.

Authors described the genetic diversity of the whitefly species of the Bemisia tabaci species complex, associated with vegetable crops in Côte d'Ivoire. In my opinion manuscript in present form is not accept to publish in Plos One. First of all in presented paper there is no answer of question why this research are important and why should be presented to international audience. Moreover, reading present paper, audience might have impression that problem is more local, not international. Authors must put more effort to the Introduction and Discusion section. I recomand the major revision of manuscript.

Introduction section - this section MUST BE improved. After reading the audience will not know why this research are important. Among others information, there is lack of goal of present work and hypothesis. Moreover I think that there is bad style of citation used in all manuscript.

L68 - Why did you use this localisation?

L104 - Why only female? Was there any differences in ratio male:female in particular localisation?

L115 - Please add some information about what kind of polimerase you used and in in which concentration?

L119 - Please add the information about electrophoresis of the PCR products

I recommend make some statistic analysis, like PCA analysis to compare the results

Discussion section: should be prepare better, there is lots of information which are not the result of conduction experiment, and at the some time lack of proper results discussion

Reviewer #2: I would like to make the following suggestions to the authors in order to improve the manuscript so that it is ready for publication:

1) Align the manuscript properly (line numbers, paragraphs, italic characters etc.). Read author guidelines carefully.

2) There are a lot of prepositions and Grammarly mistakes. Please use some relevant software to check these things or seek some help from a good English spoken person or teacher.

3) Introduction is very short. Please provide enough relevant information regarding insecticides resistant, diversity, and morphology.

4) In keywords under abstract: do not repeat the common name or scientific name of the same species (use one). Also, consider replacing with more informative and specific words e.g. genetic diversity, species diversity, Phytovirus etc. instead of vegetables, whitefly, and ecology.

5) Line:89 "If treatments were applied then how did you differentiate between the treated or untreated species? Did you take that treated species also into the account. if yes then explain".

6) Line:71-73 "it is irrelevant to mention the environmental conditions. please cite some suitable references for this or highlight these things within a figure or graph with suitable legends.

7) Line:82 "correct the spellings of whiteflies and wherever required in the whole manuscript".

8) Line 106 "Why females from site 6 or 7 were not selected"?.

9) There is misprinting in the headings of the table. make sure to correct them.

10) Line 242: Make the scientific name of the species italic and wherever required in the whole manuscript".

11) Line 255: suggest changing to “Regarding" instead of with regard to.

12) Line 335: suggest changing to "However" instead of nevertheless

13) References are very poorly written. Cite references properly. Scientific names should be italic.

Reviewer #3: In this MS., authors investigated the species diversity and their genetic diversity and structuring on smaples from vegetable crops in the major tomato production areas of Cote d’Ivoire. They found four whitefly species, including MED ASL, MED Q1, SSA1 and SSA3 by using mtCOI marker. And, they further revealed MED ASL is the main species which is the main phytovirus vector in the Ivorian vegetable cropping areas. These results provide useful information for monitoring whitefly population in Cote d’Ivoire. The methods and analysis sounds good. The conclusion and discussion are also solid and nice. The writing is very nice and clear. I think the MS is ready to published.

Reviewer #4: This study investigated the distribution and biotypes of whitefly, Bemisia tabaci in Côte d'Ivoire. The MED ASL was identified as the most popular biotype on variant vegetable plants. The analyses of the data are inadequate and the figures are not clearly displayed. The language need to be modified carefully.

6. PLOS authors have the option to publish the peer review history of their article (what does this mean?). If published, this will include your full peer review and any attached files.

Reviewer #1: No

Reviewer #2: No

Reviewer #3: No

Reviewer #4: No

---

## [Author Response · Author response to Decision Letter 0]

16 Aug 2022

Editor: Thank you for your suggestions. We apologize for any inconvenience this may have caused. We have corrected the manuscript according to the PLOS ONE style requirements and hope that you will find it suitable.

Reviewer #1: Thank you. We have taken all comments and suggestions into account and made improvements to the manuscript.

Reviewer #2: Thank you. We have taken all comments and suggestions into account and made improvements to the manuscript.

Reviewer #3: Thank you for the kind words and encouragement.

Reviewer #4: Thank you for your very pertinent suggestions. We have taken into account all reviewers' corrections and suggestions and journal requirements. We hope that all these suggestions have improved the quality of the manuscript.

---

## [Editor Report · Decision Letter 1]

23 Aug 2022

PONE-D-22-15957R1Genetic diversity of whitefly species of the Bemisia tabaci Gennadius (Hemiptera: Aleyrodidae) species complex, associated with vegetable crops in Côte d'IvoirePLOS ONE

Dear Dr. N'cho,

Thank you for submitting your manuscript to PLOS ONE. After careful consideration, we feel that it has merit but does not fully meet PLOS ONE’s publication criteria as it currently stands. Therefore, we invite you to submit a revised version of the manuscript that addresses the points raised during the review process.

We look forward to receiving your revised manuscript.

Kind regards,

Yunzhou Li

Academic Editor

PLOS ONE

Journal Requirements:

Additional Editor Comments:

please answer all question in manuscript and put reference in right place,
---

## [Author Response · Author response to Decision Letter 1]

1 Oct 2022

Review Comments to the Author

Reviewer #1: Genetic diversity of whitefly species of the Bemisia tabaci (Gennadius) (Hemiptera: Aleyrodidae) species complex, associated with vegetable crops in Côte d'Ivoire.

Authors described the genetic diversity of the whitefly species of the Bemisia tabaci species complex, associated with vegetable crops in Côte d'Ivoire. In my opinion manuscript in present form is not accept to publish in Plos One. First of all in presented paper there is no answer of question why this research are important and why should be presented to international audience. Moreover, reading present paper, audience might have impression that problem is more local, not international. Authors must put more effort to the Introduction and Discusion section. I recomand the major revision of manuscript.

Introduction section - this section MUST BE improved. After reading the audience will not know why this research are important. Among others information, there is lack of goal of present work and hypothesis. Moreover I think that there is bad style of citation used in all manuscript.

Thank you for your comments, which helped to improve our manuscript. For the introduction section, we have reworded several sentences to better show the broad interest of this study. Please see lines 43-51; lines 53-58; lines 60-66 page 3 and lines 67-70 page 4. In the discussion section, improvements have been made in order to explain our results while remaining in the context of our study. Please see lines 288-290 page 14; lines 291-294, 301, 304, 306 page 15 and lines 329-332 page 16.

We have also changed the style of citation which is now in the right PLOS ONE format, we apologize for this inconvenience.

L68 - Why did you use this localisation?

Thank you for your question. We chose this localisation because in recent years Ivorian farmers of these areas have complained about the ineffectiveness of insecticides in managing the whitefly Bemisia tabaci and reported high whitefly populations together with increased viral symptoms. The whitefly infestation has pushed some farmers to abandon their vegetable crops. Up to now, no study has looked at this problem and been able to explain this upsurge of whiteflies. That is the reason why we chose in particular those high whitefly abundance sites reported. We added one sentence in the material and method section regarding to this point: “…Those sites were chosen in the main tomato growing areas of Côte d’Ivoire where high populations of whiteflies were reported...”

L104 - Why only female? Was there any differences in ratio male:female in particular localisation?

We studied the females because of their diploid status (they are derived from fertilised eggs), whereas males are parthenogenically produced by arrhenotoky, resulting in haploid unfertilized egg. That is the reason why in all sampled sites, only females were studied, but when we sex-sorted under the binocular all samples, we did not observe any bias in sex ratio. 

L115 - Please add some information about what kind of polimerase you used and in in which concentration?

Thank you for your suggestion, but we want to clarify that for this PCR we used 12.5 µl of 2x type-it master mix (Qiagen). This Master Mix contains a specific Taq DNA Polymerase adapted to SSR types of markers as specified in the material and method section. Please see line 126 page 7.

L119 - Please add the information about electrophoresis of the PCR products

Thank you. We have added this sentence in material and method. The amplified fragments (PCR products) were analysed with an automated DNA/RNA analyser QIAxcel Advanced (QIAxel ScreenGel® software; Qiagen). Please see line 131 page 7.

I recommend make some statistic analysis, like PCA analysis to compare the results.

Thank you for your comment, indeed we performed a PCA on the data and obtained similar results as for STRUCTURE (see below where we launched all individuals, giving the clustering information of structure showing the same results). According to those results, we did not include this analysis in the manuscript, but it can be added in sup data if you think it’d be worth adding it or a sentence in the results section stating that similar results were also obtained with a PCA analysis (data not shown). 

Discussion section: should be prepare better, there is lots of information which are not the result of conduction experiment, and at the sometime lack of proper results discussion

In the discussion section, improvements have been made in order to explain our results while remaining in the context of our study. Please see lines 288-290 page 14; lines 291-294, 301, 304, 306 page 15 and lines 329-332 page 16.

Reviewer #2: I would like to make the following suggestions to the authors in order to improve the manuscript so that it is ready for publication:

1) Align the manuscript properly (line numbers, paragraphs, italic characters etc.). Read author guidelines carefully.

Thank you for your suggestion. We have aligned the manuscript properly and understood the guidelines. 

2) There are a lot of prepositions and Grammarly mistakes. Please use some relevant software to check these things or seek some help from a good English spoken person or teacher.

Thank you for this suggestion and the kind words. We have reviewed the manuscript and corrected as many mistakes as possible.

3) Introduction is very short. Please provide enough relevant information regarding insecticides resistant, diversity, and morphology.

Thank you for this suggestion. We have revised the introduction and have added some necessary information. Please see line 56-58 and line 62 and line 66 page 3.

4) In keywords under abstract: do not repeat the common name or scientific name of the same species (use one). Also, consider replacing with more informative and specific words e.g. genetic diversity, species diversity, Phytovirus etc. instead of vegetables, whitefly, and ecology.

Thank you. The keywords have been modified according to your suggestion. Please see line 39 page 2.

5) Line:89 "If treatments were applied then how did you differentiate between the treated or untreated species? Did you take that treated species also into the account. if yes then explain".

Thank you for this question. We carried out this investigation to try to explain the phenomena observed in the surveyed fields (level of infestation and the presence of a particular cryptic species). There are 12 sites where all fields were treated and one site (site 13) where the okra field was not treated. According to our results, the presence of a genetic group does not depend on whether the field is treated or not. It could depend on the type of insecticide treatment used, but we have not enough data (ie number of fields) to support this hypothesis, that is the reason why we did not include any result description or drawn conclusion from these information. The most logical explanation would be the geographical location of the sampled sites.

6) Line:71-73 "it is irrelevant to mention the environmental conditions. please cite some suitable references for this or highlight these things within a figure or graph with suitable legends.

Thank you for this suggestion. Environmental conditions have been excluded from this paragraph, because these information were not recorded during our study.

7) Line:82 "correct the spellings of whiteflies and wherever required in the whole manuscript".

Thank you for your suggestion, we have corrected the spellings of whiteflies and wherever required in the whole manuscript. 

8) Line 106 "Why females from site 6 or 7 were not selected"?.

In all sampled sites, only females were studied. All collected females with a maximum of 31 per plant species and sites were incubated in 50 µl of extraction buffer. We have corrected the sentence to make it explanatory. Please see line 116-117 page 7.

9) There is misprinting in the headings of the table. make sure to correct them.

Thank you, we have corrected misprinting in the headings of the table.

10) Line 242: Make the scientific name of the species italic and wherever required in the whole manuscript".

Thank you, we have marked the scientific name of the species in italic and wherever required in the whole manuscript. Please see line 256 page 13.

11) Line 255: suggest changing to “Regarding" instead of with regard to.

Thank you for the suggestion, we have changed the sentence in the manuscript, we have changed “regard to” to “Regarding”. Please see line 270 page 14.

12) Line 335: suggest changing to "However" instead of nevertheless

Thank you for the suggestion, we have changed the sentence in the manuscript, we have changed “nevertheless” to “However”. Please see line 347 page 17.

13) References are very poorly written. Cite references properly. Scientific names should be italic.

Thank you for this observation, we have cited references properly and put Scientific names in Italic.

Reviewer #3: In this MS., authors investigated the species diversity and their genetic diversity and structuring on smaples from vegetable crops in the major tomato production areas of Cote d’Ivoire. They found four whitefly species, including MED ASL, MED Q1, SSA1 and SSA3 by using mtCOI marker. And, they further revealed MED ASL is the main species which is the main phytovirus vector in the Ivorian vegetable cropping areas. These results provide useful information for monitoring whitefly population in Cote d’Ivoire. The methods and analysis sounds good. The conclusion and discussion are also solid and nice. The writing is very nice and clear. I think the MS is ready to published.

Thank you for the kind words and encouragement

Reviewer #4: This study investigated the distribution and biotypes of whitefly, Bemisia tabaci in Côte d'Ivoire. The MED ASL was identified as the most popular biotype on variant vegetable plants. The analyses of the data are inadequate and the figures are not clearly displayed. The language need to be modified carefully.

Thank you for your very pertinent suggestions. We have taken into account all reviewers' corrections and suggestions and journal requirements. We hope that all these suggestions have improved the quality of the manuscript.

---

## [Editor Report · Decision Letter 2]

18 Oct 2022

Genetic diversity of whitefly species of the Bemisia tabaci Gennadius (Hemiptera: Aleyrodidae) species complex, associated with vegetable crops in Côte d'Ivoire

PONE-D-22-15957R2

Dear Dr. N'cho,

We’re pleased to inform you that your manuscript has been judged scientifically suitable for publication and will be formally accepted for publication once it meets all outstanding technical requirements.

Kind regards,

Yunzhou Li

Academic Editor

PLOS ONE
---

## [Editor Report · Acceptance letter]

20 Oct 2022

PONE-D-22-15957R2 

Genetic diversity of whitefly species of the *Bemisia tabaci* Gennadius (Hemiptera: Aleyrodidae) species complex, associated with vegetable crops in Côte d'Ivoire 

Dear Dr. N'cho:

I'm pleased to inform you that your manuscript has been deemed suitable for publication in PLOS ONE. Congratulations! Your manuscript is now with our production department. 

Kind regards, 

on behalf of

Dr. Yunzhou Li 

Academic Editor

PLOS ONE